# Screening and Production of Industrially Relevant Enzymes by *Bacillus paranthracis* Strain MHDS3, a Potential Probiotic

**Musundwa Locardia Tshisikhawe, Mamonokane Olga Diale** [ID]**, Adrian Mark Abrahams** *[ID] **and Mahloro Hope Serepa-Dlamini** [ID]

Department of Biotechnology and Food Technology, University of Johannesburg, Doornfontein Campus, P.O. Box 17011, Johannesburg 2028, South Africa; locardiat@gmail.com (M.L.T.); diale.olga@gmail.com (M.O.D.); hopes@uj.ac.za (M.H.S.-D.)
* Correspondence: adriana@uj.ac.za

**Abstract:** The digestive process and intestinal protein absorption are influenced by a variety of factors. Due to their numerous health advantages, including potential favorable effects on protein digestion and absorption, probiotics have gained increased attention in recent years. Probiotics can control the intestinal microflora, which in turn affects the intestinal bacteria responsible for proteolysis. Additionally, certain probiotics can release exoenzymes that aid in the digestion of proteins and others can stimulate the host's digestive protease and peptidase activity. By boosting transport and enhancing the epithelium's capacity for absorption, probiotics can also improve the absorption of tiny peptides and amino acids as well as lessen detrimental protein fermentation, which lowers the toxicity of metabolites. The present study explored the production of enzymes by *Bacillus paranthracis* strain MHDS3, a probiotic candidate isolated from *Pellaea calomelanos. Bacillus paranthracis* displayed enzyme activities of amylase (31,788.59 IU), cellulase (4487.486 IU), and pectinase (13.98986 IU) through submerged fermentation. The CAZyme analysis of *B. paranthracis* revealed 16 CAZyme gene clusters associated with cellulose, amylase, and pectinase activity. Thus, *B. paranthracis* is a promising probiotic strain that can produce enzymes with biotechnological applications.

**Keywords:** probiotics; *Bacillus paranthracis*; enzyme activity; *Pellaea calomelanos*; submerged fermentation; digestive enzymes

## 1. Introduction

Probiotics are described as living microorganisms that when administered in sufficient concentrations, provide health advantages that benefit the host [1]. Every healthy digestive tract has a delicate balance [2]. A digestive system might become imbalanced because of illness, antibiotic use without taking probiotics, poor diet, or overgrowth of pathogenic bacteria [3]. Such an imbalance may lead to gastrointestinal infections and other kinds of infections [4]. Probiotics offer a natural way to treat gastrointestinal diseases like diarrhea and irritable bowel syndrome [5]. For a microorganism to qualify as a probiotic candidate, it must possess genotypic characteristics associated with tolerance to gastrointestinal stress and adhesion, tolerance to bile salts and gastric juices, as well as the ability to survive in an acidic environment [6].

Probiotics play a critical role in humans and animals, and some of the benefits include (a) metabolizing undigested carbohydrates [7,8], (b) preventing pathogenic bacteria and viruses from multiplying [9], (c) strengthening the host immune system and lessening of allergies and inflammation [10], (d) synthesis of various bio-active compounds [11], (e) nutrient bio-availability [12], and (f) making the environment less desirable for harmful microorganisms by changing the pH and reducing oxygen availability in the intestines [13].

Probiotics secrete important bioactive compounds that are crucial for the well-being of humans and animals [11]. Bacteriocins, enzymes, vitamins, amino acids, oligosaccharides, exopolysaccharides, short-chain fatty acids, and immunomodulatory compounds

are among the bioactive compounds produced by probiotic strains [14]. Vitamins aid with energy and amino acid metabolism, thus promoting good health [15], enzymes speed up metabolism [16], bacteriocins combat pathogens [17], and immunomodulatory molecules modulate the immune system of the host [18]. *Lactobacillus, Bifidobacterium, Propionibacterium, Streptococcus,* and certain *Saccharomyces* species are the most used probiotics genera [16,19] with reported common species such as *Saccharomyces boulardii, Lactobacillus acidophilus, Lactobacillus rhamnosus, Lactobacillus casei, Lactobacillus plantarum, Bifidobacterium longum,* and *Bifidobacterium bifidum* [20]. Probiotics are one of the disease control strategies to improve both animal and human microflora and assist by reducing gastrointestinal disorders [21]. Probiotic species such as *Lactobacillus, Bifidobacteria, Streptococcus,* and some *Saccharomyces* species are known for their long history of safe use in biotechnology, health, and food-related industries [22]. Although these species show outstanding probiotic properties, they barely survive in extremely harsh environments such as those created by acidic gastric juice and alkaline bile in the gastrointestinal tract [23]. Spore-forming bacteria such as *Bacillus* spp. can survive extreme environmental conditions [7,24]. Spores enable bacteria to withstand extreme conditions, maintain stability during heat processing, and sustain viability in low-temperature storage using different mechanisms such as dehydration of spore core, DNA protection by small soluble acids molecules, large depot of calcium dipicolinate in the protoplast, and DNA repair mechanism [25,26]. Moreover, these species are known to maintain stability in high and low-temperature storage [25]. Similarly, *Bacillus* spp. have high antagonistic activity resulting from the secretion of antimicrobial compounds such as coagulin, amicoumacin, and subtilisin, which confer probiotic benefits by inhibiting the growth of competing pathogenic microbes [26]. *Bacillus* probiotic species such as *Bacillus subtilis, B. amyloliquefaciens,* and *B. licheniformis* are well known for their ability to produce high yields of extracellular amylases, glucoamylases, proteases, cellulases, xylanases, pectinases, and lipases in their vegetative form, which improve nutrient digestion and absorption in the gut [27,28]. This makes *Bacillus* species excellent probiotics, which necessitates more studies to investigate the *Bacillus* species as a potential probiotic. In a previous study by Diale [29], the probiotic potential of *B. paranthracis* strain MHSD3 through whole genome in silico analysis was investigated, supported by in vitro assays. Considering the advantages of *Bacillus* strains and the ability to produce extracellular enzymes such as cellulase, amylase, xylanase, and phytase, it would be intriguing to find out whether *Bacillus paranthracis* MHSD3 can combine probiotic features with enzyme production. The use of microbial enzymes in food industries, pharmaceuticals, fabrics, papers, and other industries is high and growing rapidly due to their conveniences and eco-friendliness, apart from their importance in a variety of biotechnological processes, their rapid growth rate, and their ease of nutritional needs compared to enzymes from other sources such as those plant- and animal-based [30]. Prior to considering process optimization for large-scale enzyme production, it is necessary to confirm the strain's ability to produce the desired enzyme and find a suitable approach to evaluate the enzyme activity. In this study, enzymes, namely, amylase, cellulase, and pectinase were screened and produced. In addition, various kinetic characterization parameters of the enzymes were investigated, which included incubation period and resistance to environmental attributes such as pH and salinity.

## 2. Materials and Methods

### 2.1. Bacterial Strain and DNA Extraction

*Bacillus paranthracis* strain MHSD3 was previously isolated and identified initially by Mahlangu and Serepa-Dlamini [31], and the strain's identity was further confirmed through whole genome sequence analysis by Diale and colleagues [29]. Thirty percent glycerol stock cultures of the strain were preserved at −80 °C. In the current study, the glycerol stock cultures were re-sub-cultured on nutrient agar (NA) plates and incubated for 24–48 h (h) at 30 °C, and 30% glycerol stock cultures were prepared for long-term preservation at −80 °C.

## 2.2. Enzyme Screening

### 2.2.1. Amylase

The methodology used was described by Hankin and Anagnostakis [32] with minor modifications. Briefly, *B. paranthracis* strain MHSD3 was streaked on NA plates supplemented with 2% soluble starch, pH 6.0. The NA plates were incubated at 30 °C for 24 h. Following incubation, the plates were flooded with 1% iodine for 20 min (min), and clear halo zones around the colonies confirmed the presence or activity of amylase.

### 2.2.2. Cellulase

The secretion of cellulase enzyme was determined following a method described by Kim [33]. Strain MHSD3 was spread inoculated on carboxymethyl cellulose (CMC) agar media containing (g/L) (5 g peptone, 5 g yeast extract, 1 g $K_2HPO_4$, 0.2 g $MgSO_4.7H_2O$, 5 g NaCl, 10 g carboxymethyl cellulose powder, and 15 g agar (pH 7.0)) and incubated at 30 °C for 24 h. Following incubation, the plates were flooded with 0.1% Congo red for 15 min and washed with 1 M NaCl. A translucent zone around the colonies indicated cellulase secretion.

### 2.2.3. Pectinase

Pectinase screening agar media (PSAM) with the following composition (g/L) (1 g $NaNO_3$, 0.5 g $MgSO_4$, 1 g $K_2HPO_4$, 0.5 g yeast extract, 20 g agar, and 10 g citrus pectin (Sigma-Aldrich), pH 7.0) was used to screen for pectinase production following the method by Mohandas [34]. Incubation was at 30 °C for 48 h, then the plates were flooded with 50 mM iodine potassium iodide solution for 5 min with gentle agitation. The presence of translucent halo zones around the colonies indicated pectinase production.

## 2.3. Optimizations of Different Parameters for Increased Enzyme Yield

Following a successful screening of the above-mentioned enzymes, different growth conditions were optimized for increased enzyme production.

### 2.3.1. pH

The enzyme production was carried out at different pH from low to high based on the pH requirement for a specific enzyme. Flasks were incubated at 30 °C, agitating at 150–200 rpm for an optimized incubation period. Enzyme activity was assayed as described in Section 2.4.

### 2.3.2. Incubation Period

To determine the effect of the incubation period on enzyme production, strain MHSD3 was inoculated in respective media for each enzyme and incubated at 30 °C for 7 days, agitating at 150–200 rpm. Samples were aliquoted every 24 h and enzyme activity was assayed as described in Section 2.4.

### 2.3.3. Temperature

The effect of temperature on enzyme production was determined by inoculating strain MHSD3 in respective media and incubated at various temperatures (30, 40, 50, and 60 °C). Enzyme activity was assayed as described in Section 2.4.

### 2.3.4. Carbon and Nitrogen Sources

The effect of carbon and nitrogen sources on the enzyme production was determined by supplementing the media with carbon and nitrogen sources as per Table 1, and media were adjusted to optimum pH, temperature, and incubation period.

**Table 1.** Carbon and nitrogen source effect on enzyme production.

| Substrate | Amylase | Cellulase | Pectinase |
|---|---|---|---|
| **Carbon source** | 1%<br>Glucose<br>Corn steep<br>Starch<br>Galactose<br>Sucrose | 1%<br>Dextrose<br>Mannitol<br>Starch<br>Sucrose | 1%<br>Glucose<br>Sucrose<br>Casein |
| **Nitrogen source** | 0.5%<br>Peptone<br>Beef extract<br>Yeast extract<br>Ammonium chloride | 1%<br>Peptone<br>Yeast extract<br>Urea<br>Ammonium sulfate | 1%<br>Urea<br>Ammonium chloride<br>Peptone<br>Yeast extract |

### 2.3.5. Effect of NaCl

The effect of salinity on enzyme activity was investigated by measuring enzyme activity under standard assay conditions at varied NaCl concentrations (0, 5, and 7.5%) and the enzyme activity was assayed as described in Section 2.4.

### 2.4. Quantitative Analysis of Enzyme Production

Enzyme production was quantitatively confirmed for all enzymes screened. Twenty-four hours bacterial culture of strain MHSD3 was inoculated in 100 mL Luria Bertani (LB) broth and incubated at 30 °C agitating at 150 rpm. The 24 h culture was used for quantitative analysis of the enzymes.

### 2.4.1. Amylase Production and Activity Assay

The method by Abd-Elhalem et al. [35] was adopted with modifications. A hundred milliliters of media (g/L) (10 g soluble starch, 5 g yeast extract, 0.5 g $KNO_3$, 1 g $MgSO_4$ $7H_2O$, 1 g $KH_2PO_4$, and 0.1 g $CaCl_2$ $2H_2O$, pH 5.6) was inoculated with 2.5 mL (*v/v*) of 24 h culture and incubated at 30 °C agitating at 200 rpm for 48 h. Following incubation, the media was centrifuged for 10 min at 10,000 rpm at 4 °C, and the cell-free supernatant was used for enzyme activity assay.

Amylase activity was determined following the method by Simair et al. [36] and Miller [37] with slight modifications. Briefly, 0.5 mL of 1% soluble starch was added to 0.5 mL cell-free supernatant and incubated for 10 min at 37 °C. The reaction was stopped by adding 2 mL of 3,5-dinitrosalicylic acid (DNS) reagent and placed in a boiling water bath for 5 min. The absorbance was measured at 540 nm using a spectrophotometer (Biomate 3, Thermo Fisher Scientific, Waltham, MA, USA) against a blank. International unit for enzymes (IU) of amylase was defined as the amount of enzyme that could hydrolyze starch and release 1 µmol of reducing sugar (glucose) per min under assay conditions.

### 2.4.2. Cellulase Production and Activity Assay

A method described by Temsaah et al. [38] was adopted with modifications. A hundred milliliters of media (g/L) (20 g $NaNO_3$, 12 g $K_2HPO_4$, 1 g $KH_2PO_4$, 5 g KCl, 5 g $MgSO_4.7H_2O$, 2 g yeast extract, 1 g $FeSO_4.7H_2O$, 10 g carboxymethyl cellulose (CMC) pH 8) was inoculated with 2.5 mL v/v of 24 h culture and incubated at 30 °C agitating at 150 rpm for 48 h. Following incubation, the media was centrifuged for 20 min at 10,000 rpm at 4 °C, and the cell-free supernatant was used for enzyme activity assay.

Cellulase activity was determined following the method by Islam and Roy [39] and Miller [37] with slight modifications. One percent of CMC in 1 M citrate-phosphate buffer (pH 5.4) was considered as a substrate. Briefly, 100 µL of cell-free supernatant was added to 1 mL of CMC solution and 1 mL citrate buffer pH 9 and incubated for 60 min at 40 °C. The reaction was halted by adding 3 mL of 3,5-dinitrosalicylic acid (DNS) reagent and placed in a boiling water bath for 10 min. The absorbance was measured at 540 nm using a

spectrophotometer (Biomate 3, Thermo Fisher Scientific) against a blank without enzyme. IU of cellulase activity was defined as the amount of enzyme that could hydrolyze CMC and release 1 μmol of glucose within 1 min under assay conditions.

### 2.4.3. Pectinase Production and Activity Assay

A method described by Mohandas et al. [34] was adopted with modifications. A hundred milliliter of media (g/L) (1 g NaNO$_3$, 0.5 g MgSO$_4$, 1 g K$_2$HPO$_4$, 0.5 g yeast extract, and 10 g citrus pectin pH (7)) was inoculated with 2.5 mL (*v/v*) of 24 h culture and incubated at 30 °C, agitating at 150 rpm for 48 h. Following incubation, the media was centrifuged for 10 min at 10,000 rpm at 4 °C, and the cell-free supernatant was used for enzyme activity assay. Pectinase activity was determined following the method by Mohandas et al. [34] and Miller [37] with modifications. Briefly, 0.5 mL of cell-free supernatant was added to 0.5 mL of 1% citrus pectin in 0.1 M acetate buffer, pH 6.0, and incubated for 10 min at 40 °C. The reaction was stopped by adding 1.5 mL of 3,5-dinitrosalicylic acid (DNS) reagent and placed in a boiling water bath for 5 min. The absorbance was measured at 540 nm using a spectrophotometer (Thermo Fisher Scientific) against a blank without enzyme. IU of pectinase was defined as the amount of enzyme that liberated 1 μmol monogalacturonic acid per min under assay conditions.

### 2.5. Identification of Genes Involved in Amylase, Pectinase and Cellulase Activity

To investigate pectinase, amylase, and cellulase genes, the draft genome of *B. paranthracis* MHSD3 (JABGBK000000000) was sequenced, assembled, and annotated using the National Center for Biotechnology Information—Prokaryotic Genome Annotation pipeline (PGAP) as described by Diale et al. [29]. To further identify carbohydrate-active enzymes (CaZyme) genes, the genome was annotated using dbCAN3 meta server (https://bcb.unl.edu/dbCAN2) (Accessed 16 July 2023) following default settings with dbCAN (HMMER), CaZy (using Diamond), and HMMER (dbCAN-sub) [40].

## 3. Results and Discussion

Probiotics aid in digestion through regulating the intestinal microbiota and altering gut microbes in proteolysis. Probiotics can also stimulate host digestive amylase and peptidase activity in the host, and some can release exoenzymes that assist in protein digestion [41]. Furthermore, probiotics can promote small peptide and amino acid absorption by improving the absorption ability of epithelium and reducing detrimental protein fermentation, thus lowering the toxicity of metabolites [41].

### 3.1. Primary Screening of Each Enzyme

*Bacillus paranthracis* strain MHSD3 was grown in primary screening plates with suitable substrate to induce enzyme production. The strain showed amylase, cellulase, and pectinase production. Halo zones around the colonies and color change were indicative of positive results (Figure 1). Halo zones around the colony indicating production of amylase (Figure 1a), cellulase (Figure 1b), and pectinase (Figure 1c) screening.

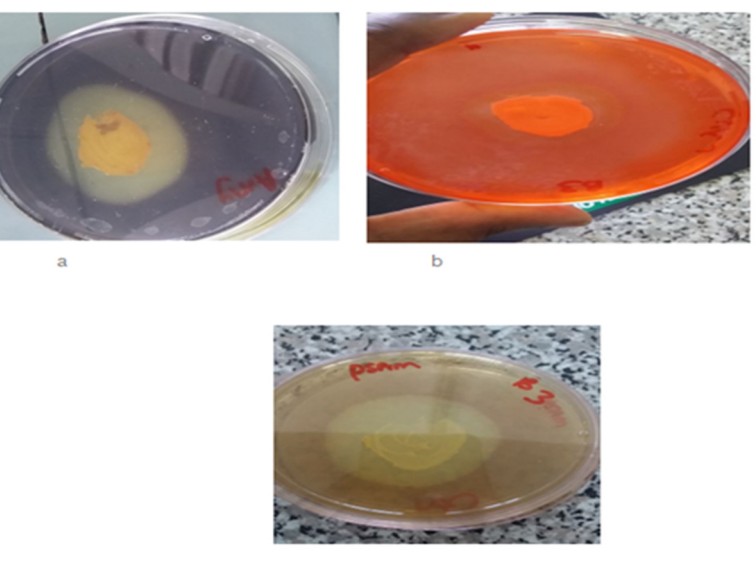

**Figure 1.** Halo zones around the colony indicating production of (**a**) amylase, (**b**) cellulase, and (**c**) pectinase screening.

*3.2. Quantitative Screening*

3.2.1. Amylase Production and Characterization

Effect of pH

The production of amylase was reported at pH 5.6 with 22,067 IU (Figure 2A). Enzyme production in *Bacillus* strains has been reported at pH levels ranging from 6 to 7. When the initial pH of the experiment was at pH 5.6, a high enzyme titer was obtained (Figure 2A). This study revealed that the amylase enzyme produced by *B. paranthracis* can be produced successfully in a wide range of pH settings, ranging from slightly acidic to slightly alkaline. Similar studies were recorded at pH 6.0, where *B. licheniformis* HULUB1 and *B. subtilis* SUNGB2 produced the highest amounts of amylase 0.261 mg/mL and 0.154 mg/mL, respectively [42]. The highest enzymatic production of 15.89 U/mL was recorded for probiotic *Lactobacillus plantarum* 445 at pH 7 [43]. Abd-Elaziz et al. [44] reported the highest enzyme production of 32.5 U/mL for *Bacillus atrophaeus* NRC1 at pH 6, while Rakaz et al. [45] reported the highest enzyme production of *B. cereus* (247.20 U/mL) and *B. licheniformis* (15.959 U/mL) at pH 8. In another study *Bacillus amyloliquefaciens* P-001 has the optimal enzyme production at pH 9 [46]. It was also noticeable that at extreme acidic and alkaline conditions, amylase activity was at its lowest. This could be attributed to the concentration of hydrogen ions present in solution as reported in the literature [47]. Microorganisms are sensitive to the concentration of hydrogen ions present in the media; therefore, pH is one of the most critical parameters that determines their growth and enzyme secretion [47].

Effect of Incubation Period

The broths were incubated for a period of 168 h to ascertain when maximum enzymatic activity was achieved. The highest enzymatic production was achieved at 48 h with an activity of 28,711 IU (Figure 2B). The enzyme production increased as the incubation period was increased until the optimum period was obtained. In most cases, as the incubation time increased, the enzyme production declined. This could be due to nutrient depletion in the media as reported [48]. Similar results were reported by Yassin et al. [49] where the highest enzyme titer was produced at 48 h of incubation, which is congruent with this study. While Rakaz et al. [45] reported the highest enzyme production at 24 h for *Bacillus* sp.

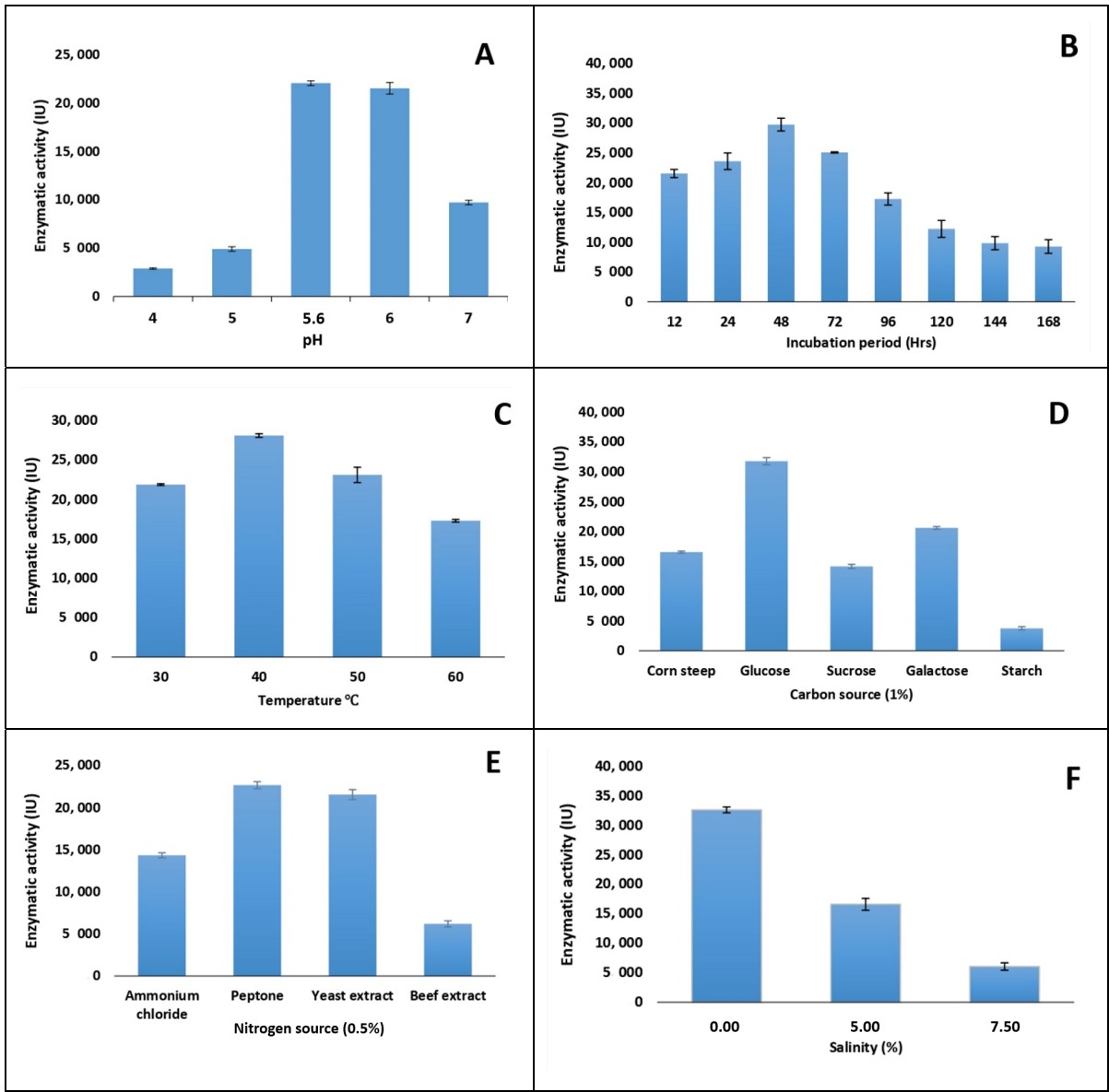

**Figure 2.** (**A**) The effect of pH, (**B**) incubation period, (**C**) temperature, (**D**) carbon source, (**E**) nitrogen source, and (**F**) salinity on amylase production by *B. paranthracis* MHSD3.

Effect of Temperature

The effect of temperature on amylase production was investigated for 48 h at four different temperatures (30, 40, 50, and 60 °C). Amylase production was observed in all temperatures. At 40 °C, maximum amylase production was observed with an enzyme activity of 28,110 IU (Figure 2C). As the temperature increased further, amylase titer decreased. Higher temperatures denature proteins, carrier proteins, and enzymes, which lead to cell death [50]. Similarly, Siroosi et al. [51] and Dike et al. [52] reported maximum amylase production from haloarchaea strain D61 and *B. circulans* at 40 °C and 35 °C, respectively, whereas Simair et al. [36] reported maximum amylase production at higher temperature (50 °C) from *Bacillus* sp. BCC 01-50.

Effect of Carbon and Nitrogen Sources

Different carbon sources, such as glucose, starch, corn steep, galactose, and sucrose, were investigated for the most suitable source for amylase production. The carbon sources have an immense impact on amylase production. Starch and sucrose showed low amylase

production. Glucose had the highest amylase production at 31,788.59 IU followed by galactose (Figure 2D). The results in this study correspond with the findings of *Bacillus cereus* MTCC 1305 [53], *Bacillus megatherium* [54], and *Penicillium* sp. SP2 [55,56], which demonstrated high production of amylase when glucose was compared to other carbon sources. Contrary to the findings of this study, other studies reported maximum amylase production when starch was used as a carbon source for *Bacillus licheniformis* ZB-05 [57], *Cronobacter sakazakii Jor* 52 [58], and *Bacillus* sp. [59]. Nitrogen source optimization was carried out in media containing 1% glucose and 0.5% (peptone, yeast extract, beef extract, and ammonium chloride), separately. Peptone had maximum amylase production with enzyme activity of 22,668 IU (Figure 2E) compared to other nitrogen sources. Similarly, Simair et al. [36], Acharya et al. [60], Aladejana et al. [61], and Khushk et al. [62] reported maximum amylase production when peptone was used as a nitrogen source. Contrary to this study, beef extract had the maximum production for *Bacillus* sp. BCC 01-50 [36]. In another study where *Bacillus subtilis* D19 was the source, maximum production of amylase was reported when yeast extract was used as a nitrogen source [63].

Sodium chloride concentration may either accelerate or inhibit the rate of an enzyme reaction. The alteration of salinity levels can impact the habitat of microorganisms, which is commonly recognized as the source of enzymes. In addition, it affects enzyme activities by denaturing proteins and reducing their solubility. Salts remove the essential layer of water molecules from the protein surface, eventually denaturing the protein. The highest enzyme activity was observed with no NaCl addition at 32,576.74 IU, and 5% NaCl had 16,588.43 IU activity (Figure 2F). The enzyme activity was reduced by a high concentration of NaCl. Conversely, NaCl served as an inhibitor of amylase of *Aspergillus niger* ATCC 1004 [64].

### 3.2.2. Cellulase Production and Characterization

### Effect of pH on Cellulase Production

Enzyme production was observed between pH 6 and 9, and the maximum enzyme production was at pH 8 at 2021 IU (Figure 3A). Cellulase enzyme production was observed from slightly acidic to alkaline conditions. Similar findings were observed in *Bacillus* sp. PM06 [65]. In another study, maximum production of cellulase was observed at pH 9 by *Bacillus* sp. [66]. In contrast, Islam et al. [67] observed maximum production of cellulase at pH 3.5 by *Bacillus* sp.

### Effect of the Incubation Period

The incubation period is one of the most crucial factors in the production of enzymes, and maximum enzyme production is only possible after a certain incubation period [68]. Cellulase production gradually increased from 24 to 48 h, then declined more rapidly as the fermentation progressed (Figure 3B). At 48 h of cultivation, the maximal enzyme production was observed with an activity of 3983 IU (Figure 3B). A similar outcome was discovered in *Bacillus* sp. PM06 [65]. Maximum cellulase production was reached within 72 h for both *Streptomyces* sp. BRC1 and *Streptomyces* sp. BRC2 [69]. In contrast, *Bacillus* sp. had maximum cellulase production after 24 h [67].

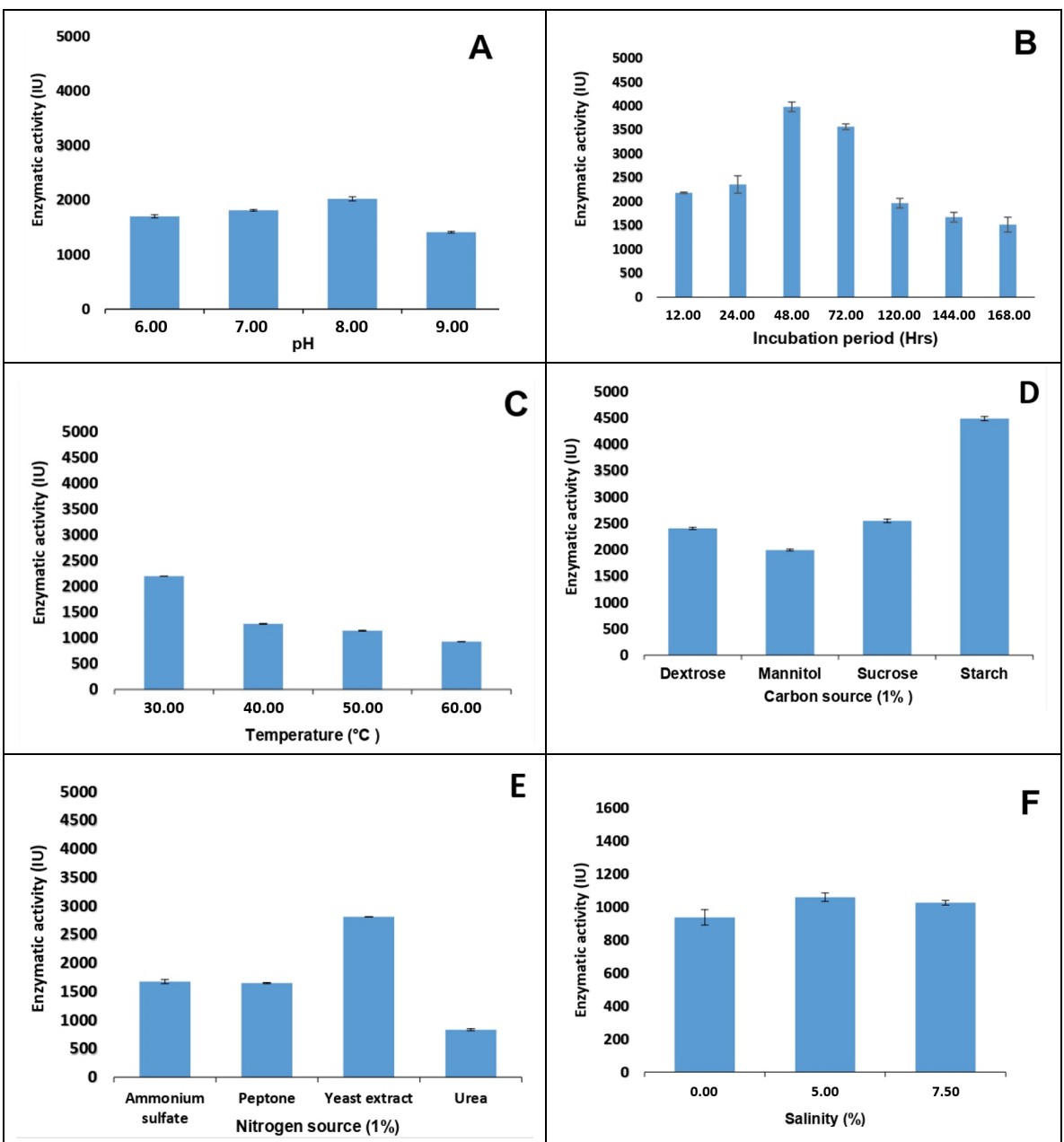

**Figure 3.** (**A**) The effect of pH, (**B**) incubation period, (**C**) temperature, (**D**) carbon source, (**E**) nitrogen source, and (**F**) salinity on Cellulase production by *B. paranthracis* strain MHSD3.

Effect of Temperature on Cellulase Production

Temperature is also an important factor in the production of an enzyme [70]. The rate of microbial growth, enzyme production, enzyme inhibition, and protein denaturation are all affected by temperature [71]. *Bacillus paranthracis* strain MHSD3 showed maximum cellulase production at 30 °C, with an enzymatic activity of 2199 IU (Figure 3C). Cellulase production was lowered as the temperature increased. Similar findings were reported for *Aspergillus niger* [72]. Maximum production was reported at 29 °C for *Trichoderma reesei* QM 9414 [73]. *Bacillus subtilis* and *B. circulans* produced minimum cellulase at 45 °C, whereas the maximum production was obtained at 40 °C [74].

Effect of Carbon and Nitrogen Source

In most microbial fermentation processes, carbohydrates are used as carbon sources that stimulate growth, resulting in the production of primary metabolites like enzymes [75]. Nitrogen sources are used to enhance enzyme production. Most commercial enzymes use nitrogen sources that are either organic or inorganic or both. Growth is usually accelerated when organic and inorganic nitrogen sources are available [75].

The carboxymethyl cellulose was replaced with other carbon sources such as starch, dextrose, mannitol, and sucrose. After 48 h, maximum production of cellulase was observed when starch was used as a carbon source with the enzymatic activity of 4487 IU (Figure 3D). Similar findings were reported for *Bacillus* sp. PM06 [65]. In contrast, *Pseudomonas fluorescens* and *Bacillus subtilis* had maximum cellulase production when glucose was used as a carbon source [76]. In the growth medium, the influence of nitrogen sources was investigated by replacing peptone with ammonium sulfate, urea, and yeast extract. Yeast extract was found to be the best nitrogen source for cellulase production among other nitrogen sources evaluated, with an enzymatic activity of 2807 IU (Figure 3E). Similar findings were reported for *Bacillus* sp. C1AC55.07 [77]. Contrary to these studies' ammonium nitrate was found to be a suitable nitrogen source for *Bacillus* sp. [78].

Effect of Salinity on Cellulase Production

Sodium chloride did not have much effect on the activity of cellulase, optimum activity was observed with the addition of 5% NaCl with the activity of 1059.00 IU (Figure 3F).

3.2.3. Pectinase Production and Characterization

Effect of pH

As an influence of enzyme synthesis, the initial pH of the fermentation medium is critical. The microbial enzyme stability is also influenced by the medium's hydrogen ion concentration. High pectinase production was observed in this investigation when the pH was adjusted to 7.0 with the activity of 14,46 IU (Figure 4A). This could be because the optimum pH to produce pectinase is more closely related to the optimum conditions required for the growth of the specific microorganism [79]. These results agree with the production of pectinase by *Bacillus sphaericus* (MTCC 7542) at pH 6.8, which was the original pH of the media [80]. Oumer and Abate [81] reported high pectinase at an initial pH 6.5 by *Bacillus subtilis* strain Btk 27. Contrary results reported high production at acidic pH by *Bacillus* sp. MBRL576 [82].

Effect of Incubation Period

The period of incubation has a significant impact on the development of microbial products [83]. The amount of enzyme produced varied depending on the incubation period. The pectinase production constantly increased in this study up to 48 h of incubation followed by a decline after 48 h. As a result, the optimal time for pectinase production was 48 h with the enzymatic activity of 9.98 IU (Figure 4B). Similar findings were reported by Oumer and Abate [81] and Berutu et al. [84] with high pectinase production at 48 h. Martin and Morata de Ambrosini [85] reported high pectinase production after 24 h, which is contrary to the findings of this study.

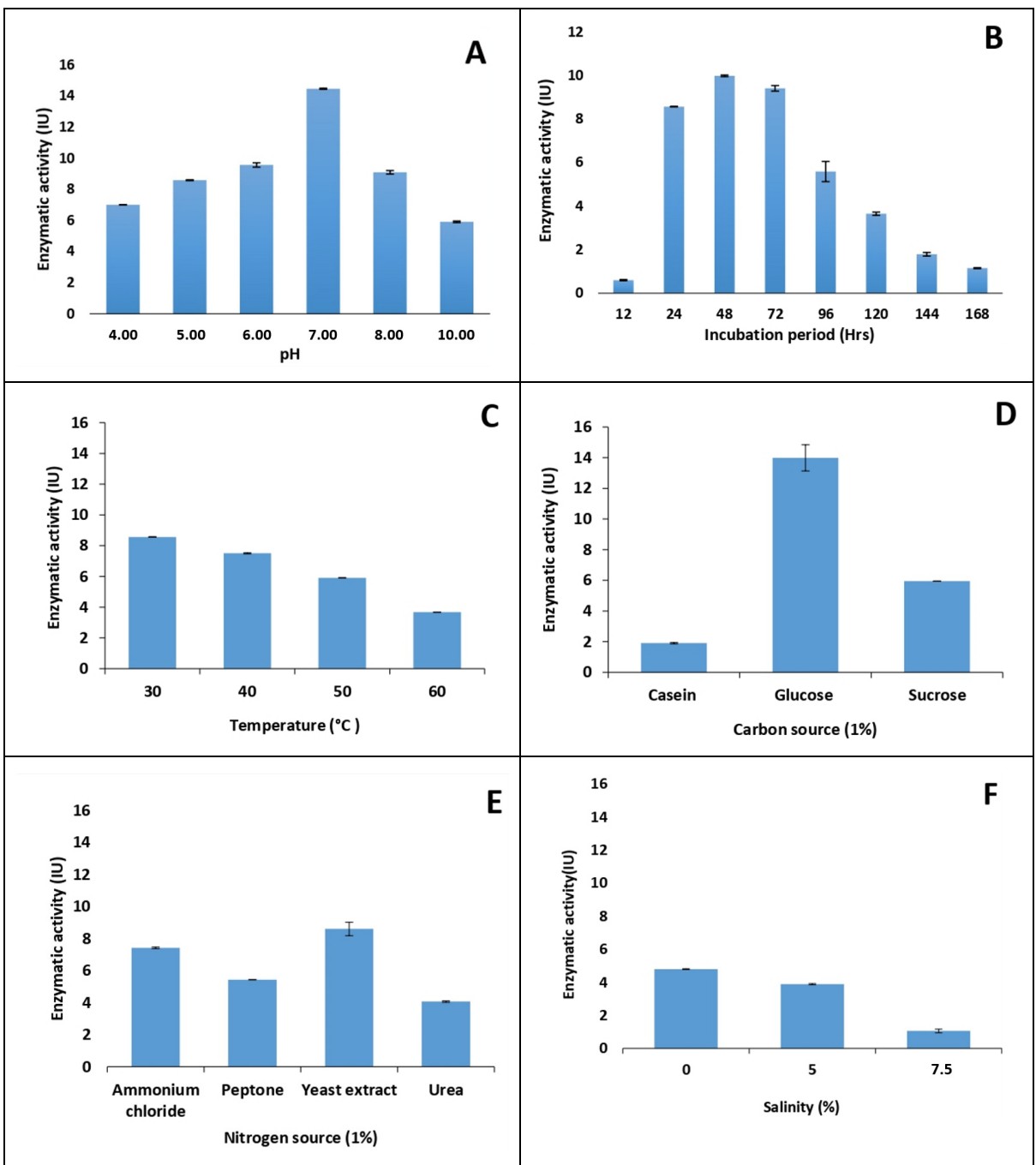

**Figure 4.** (**A**) The effect of pH, (**B**) incubation period, (**C**) temperature, (**D**) carbon source, (**E**) nitrogen source, and (**F**) salinity on pectinase production by *B. paranthracis* strain MHSD3.

Effect of Temperature

Temperature is an important factor in microbial growth as well as in the production of microbial products. The rate of microbial growth, enzyme production, enzyme inhibition, and protein denaturation are all affected by temperature [86]. In this study, enzyme production was observed at temperatures ranging from 30 to 60 °C with pectinase maximum production at 30 °C with the enzymatic activity of 8.56 UI (Figure 4C). The findings correspond with those by Tepe and Dursun [87], which showed the ideal temperature for pectinase production by *Bacillus pumilus* at 30 °C; Ibrahim et al. [88] and Thakur et al. [89] also recorded similar findings for *Aspergillus niger* HFD5A-1 and for *Bacillus tropicus* at 37 °C, respectively.

#### Effect of Carbon Source and Nitrogen Source

An appropriate supply of carbon as an energy source is essential for optimum growth, which affects the organism's growth. Pectinase production was increased when glucose was used as a carbon source with the enzymatic activity of 13.98 IU (Figure 4D), which is similar to the results of different researchers who observed maximum pectinase production when glucose was used as a carbon source [90–92]. When glucose, lactose, and xylose were added to the media, the production of pectinase decreased marginally using *Bacillus subtillis* SAV-21 [93]. In contrast to our results, El-Shishtawy et al. [54] reported repression of pectinase when glucose, lactose, maltose, and starch were used as sole carbon sources. A study in which orange peel was used as a carbon source reported the highest titer of pectinase [94].

Maximum production of pectinase was observed when yeast extract was used as a nitrogen source with an enzymatic activity of 8.60 IU (Figure 4E). The findings are similar to the study in which yeast extract was the best nitrogen source for pectinase production by *B. licheniformis* [95]. Previous studies reported the highest production when peptone was used as a nitrogen source by *Bacillus subtillis* [90]. In another study, ammonium sulfate was reported to be the best nitrogen source for pectinase production by *Bacillus* sp. [93].

#### Effect of Salinity on Pectinase production

Maximal enzyme activity was observed with no NaCl and had an activity of 4.80 IU (Figure 4F); the addition of NaCl lowered the activity of the enzyme. To be considered a potential probiotic, bacteria must possess several desirable characteristics, such as the ability to survive in a low pH environment and adhere to the site of action in a physiologically functioning state [27].

In a previous study, strain MHSD3 demonstrated in vitro probiotic candidate traits such as bile salt and gastric juice tolerance [29]. In addition, it exhibited strong cell surface characteristics such as auto-aggregation and hydrophobicity [29]. The current study showed that strain MHSD3 produced cellulase and pectinase enzymes. These are enzymes that are mostly used in animal feeds for digestion and absorption promotion [96]. Padmavathi et al. [16] reported the production of amylase (1253.4 IU/mg), lipase, and protease by *Lactobacillus* sp. G3_4_1TO2, a potential probiotic lactic acid bacterium. *Lactobacillus* and *Bifidobacteria* spp. mostly produce amylase and lipase enzymes [15]. Moreover, *Bacillus* sp. has previously been shown to be an important species due to its ability to produce a variety of extracellular hydrolytic enzymes [97].

#### Pectinase, Cellulose, and Amylase Genes Identified from *B. paranthracis* MHSD3

The PGAP annotation revealed two genes encoding for cellulose production (*celB* and *celF*) and one alpha-amylase gene (*amyA*). Based on the CAZy annotation, five genes related to amylase activity were identified, which include GH13, GH26, CBM34, CBM41, and CBM48 genes, and six GH13 subfamilies (Table 2). The GH13, also known as the alpha amylase family, is the largest family of glycoside hydrolases, representing a varied number of enzymes such as α-amylases, α-glucosidases, α-1,4-glucan branching enzymes, pullulanases, cyclodextrin glucanotransferases, and 4-α-glucanotransferases andoligo-α-1,6-glucosidases [98]. Amylase (E.C.3.2.1.1)-producing probiotics play an important role in the digestive tract of animals and human beings [16]. In addition, the *B. paranthracis* strain MHSD3 genome contains five pectinase genes and four cellulose genes (Table 2). Some of the pectinolytic enzymes identified include endo-polygalacturonase and polygalacturonase, rhamnosidase, and arabinofuranosidases, which are associated with the breakdown of pectin [99]. Pectinolytic and cellulose enzymes have been reported to be used in the production of animal feed, and in the pharmaceutical and beverage industries [100–102]. Taheri et al. [103] stated that a probiotic strain that has enzymatic activities can improve digestion, especially in newly hatched chicks. In addition, enzymes produced by probiotic strains such as *Bacillus coagulans* have been reported to degrade proteins and carbohydrates into smaller peptide molecules and free amino acids, thereby promoting metabolism in the

upper part of the small intestine, and improving the intestinal environment of the colon, and reducing toxic metabolites [24].

**Table 2.** CAZyme genes identified from *Bacillus paranthracis* MHSD3 genome.

| Gene ID/Contig Number | Enzyme Activity | Family Group |
|---|---|---|
| **Genes encoding pectinolytic enzymes** | | |
| contig_4_56 | Endo-polygalacturonase (EC 3.2.1.15) | GH28 |
| contig_1_502 contig_3_223 | Rhamnogalacturonase (EC 3.2.1.171), Xylogalacturonan hydrolase (EC 3.2.1.174), Polygalacturonase (EC3.2.1.15) | GH28 |
| contig_1_362 | α-L-rhamnosidase (EC 3.2.1.40) | GH78 |
| contig_4_458 | α-L-fucosidase (EC 3.2.1.51) | GH95 |
| contig_12_94 | Feruloyl esterase (EC 3.1.1.73) | CE1 |
| contig_5_162 | Binding to galactose, lactose, polygalacturonic acid, and LacNAc | CBM32 |
| **Amylase genes** | | |
| contig_22_33 contig_4_56 | α-amylase (EC 3.2.1.1) | GH13 CBM34 |
| contig_2_420 | α-amylase (EC 3.2.1.1), maltogenic amylase (EC 3.2.1.133), maltotetraose-forming α-amylase (EC 3.2.1.60), isoamylase (EC 3.2.1.68), maltohexaose-producing α-amylase (EC 3.2.1.98) | GH13 |
| contig_10_170 | Pullulanase (EC 3.2.1.41) | GH13_14 |
| contig_16_5 | Cyclomaltodextrinase (EC 3.2.1.54), glucan 1,4-alpha-maltohydrolase (EC 3.2.1.133), neopullulanase (EC 3.2.1.135) | GH13_20 |
| contig_11_75 | alpha,alpha-phosphotrehalase (EC 3.2.1.93) | GH13_29 |
| contig_16_6 | Glucan 1,6-a-glucosidase (EC 3.2.1.70) | GH13_31 |
| contig_7_217 | Modules from the CBM41 family bind to the α-glucans amylose, amylopectin, pullulan, and oligosaccharide fragments derived from these polysaccharides | CBM41 |
| contig_10_4 | 1,4-a-Glucan branching enzyme (EC 2.4.1.18) | GH13_9 CBM48 |

**Table 2.** *Cont.*

| Gene ID/Contig Number | Enzyme Activity | Family Group |
|---|---|---|
| Contig 5_298 | α-amylase (EC 3.2.1.1) | GH26 |
| **Cellulase genes** | | |
| contig_4_56 | Cellulase (EC 3.2.1.4) | GH18 |
| contig_3_402 | Cellulase (EC 3.2.1.4) | GH5_11 |
| contig_13_89 | cellulose synthase (EC 2.4.1.12) | GT2 |
| contig_12_98 | beta-glucosidase (EC 3.2.1.21) | GH1 |

## 4. Conclusions

*Bacillus paranthracis* strain MHSD3 is a promising probiotic strain that has the capability to produce amylase and cellulase, which are responsible for breaking down oligosaccharides into smaller molecules. The strain is a novel strain with the ability to produce enzymes with various biotechnological applications in food production, bioremediation, cosmetics, and environmental remediation, which makes microbial enzymes a preference over enzymes derived from plants and animals. Enzymes from plants and animals are costly, and an enzyme like pectinase cannot be expressed by animals; however, microbes can easily express it at a low cost. In addition, amylase enzymes facilitate the assimilation of nutrients within the digestive system. Using submerged fermentation, we statistically optimized the factors influencing the production of amylase, cellulase, and pectinase. The findings suggested the possible use of *Bacillus paranthracis* MHSD3 in animal nutrition, which can both increase the feed's digestibility and have a probiotic effect. Thus, we recommend purification of these enzymes for further characterization and application. Furthermore, in vivo studies are required to confirm strain MHSD3 as a probiotic and observe its benefits within suitable hosts.

**Author Contributions:** M.H.S.-D.: Conceptualization. M.L.T.: Data curation, writing—original draft preparation, visualization, and investigation. M.H.S.-D.: Supervision, and reviewing. A.M.A.: Calculations, validation, and reviewing. M.O.D.: Software, and writing—reviewing and editing. All authors have read and agreed to the published version of the manuscript.

**Funding:** This work was supported by the National Research Foundation (NRF) of South Africa—Thuthuka grant no. TTK210216586709

**Institutional Review Board Statement:** The study did not require ethical approval.

**Informed Consent Statement:** Not applicable.

**Data Availability Statement:** The article includes information pertaining to the tables, diagrams, and calculations.

**Conflicts of Interest:** The authors declare no conflict of interest.

## Abbreviations

| | |
|---|---|
| DNS | 3,5-dinitrosalicylic acid |
| CAZyme | Carbohydrate-active enzymes |
| PGAP | Prokaryotic Genome Annotation pipeline |
| CMC | Carboxymethyl cellulose |
| IU | International unit for enzyme |

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
