# Peer review of "Screening and Production of Industrially Relevant Enzymes by Bacillus paranthracis Strain MHDS3, a Potential Probiotic"

_fermentation, doi:10.3390/fermentation9110938_

Round 1
Reviewer 1 Report
Comments and Suggestions for Authors
Manuscript ID: fermentation-2647751
The authors of the above-mentioned manuscript report on their findings on the secretion and production of amylase, cellulase and pectinase enzymes in the strain Bacillus paranthracis MHDS3. According to the authors, the strain is a potential probiotic strain. The authors tested several culture conditions for the extracellular production of these enzymes and found that most factors, such as pH and salinity of cultivation medium, cultivation temperature, carbon and nitrogen source, had an influence on the extracellular enzyme production of amylase, cellulase and pectinase. In addition, the authors scanned the genome sequence of this strain for potential cellulase-, amylase- and pectinase-encoding genes.
The introduction is clearly written. The characteristics of a probiotic strain are stated. In order to confirm a strain being probiotic, in vivo experiments focused on, e.g., the immune system of the host, would be needed, as stated by the authors in the Conclusions. The data of the present manuscript are limited to the production of certain enzymes. The data are confined to the secretory production of cellulases, pectinases and amylases. High enzyme production levels of these enzymes can be attributes of probiotic strains; however, there are more important qualities of probiotic strains, namely, e.g., a positive effect on the host immune system, adhesion to intestinal epithelial cells and tolerance to the gastrointestinal environment, such as bile salts and acidic conditions. Some of these qualities have been determined in a previous paper.
Issues. Points to be addressed by the authors:
1. In the results and discussion section, there is no information given concerning the extracellular amylase activities reported for other strains (probiotic or not). In other words, is 22067 IU a high or a low value? The authors state in the Conclusion section that the strain is a good producer of amylases, cellulases and pectinases. If I am not mistaken, there are no data provided that warrant this statement.
2. What is the optimal growth temperature and pH of B. paranthracis MHDS3? Is this related to the pH dependence of enzyme secretion?
3. Why do the authors report amylase activity data between pH 4.0 and 7.0? Whereas they report cellulase activity data in a different pH range (pH 6.0 - 9.0) and pectinase activity between pH 4.0 and 10.0.
4. Table 2, Pectinase genes: I do not understand why rhamnosidase- and fucosidase-encoding genes are mentioned in this table under the label “Pectinase genes”. What is the meaning of this?
5. Conclusions: Why do the authors mention lipases on line 413. There are no experimental data about lipases in the text.
6. For researchers who are less familiar with probiotics, the authors could provide information concerning animal- and human-derived pectinases, amylases and cellulases. Are these enzymes derived only from the intestinal microbiota?
7. Lines 61-62: remove the parentheses.
Author Response
Thank you very much for taking the time to review our manuscript. Please see the attachment for a detailed responses and the corresponding revisions/corrections in the re-submitted files.

Reviewer 2 Report
Comments and Suggestions for Authors
The manuscript investigated the enzyme activity of a promising probiotic Bacillus paranthracis strain MHDS3. The paper is interesting and generally well-written. It is within the scope of Fermentation journal. Before publication, some issues are addressed, as follows.
1. Lines 20, 47, and 53, etc. The genus and species of microorganisms should be written in italics.
2. Lines 55-57. This description is not accurate, the majority of prebiotics work in the intestines, not the stomach, which is an acidic environment. The intestine is not in an extreme environment.
3. Lines 58-59. The reasons why Bacillus can grow in extreme environments should be elucidated.
4. Introduction. It is necessary to clearly state the purpose of the paper, which is to conduct experiments addressing which bottleneck problem exists in anaerobic digestion based on biochar additive method.
5. Line 205. minor mistakes
6. Conclusion. The majority is qualitative descriptions. Please provide some information that can be taken home.
7. There exist some English and grammar errors that need to be corrected in order to improve the quality of the manuscript.
Author Response

(The authors gave the same response as above.)

Round 2
Reviewer 1 Report
Comments and Suggestions for Authors
Lines 84-87: This is absolutely incomprehensible. What has your screening to do with the amino acid composition in the active site of the enzymes?
Table 2: Instead of pectinase genes, it would be maybe better to use a term such as Genes encoding pectinolytic enyzmes.
Comments on the Quality of English Language
Lines 84-87: This is absolutely incomprehensible.
Reviewer 2 Report
Comments and Suggestions for Authors
The author has well addressed the existing issues, I think the revised manuscript can be accepted.
